# A Target Detection Method of Distributed Passive Radar without Direct-Path Signal

**Huijie Zhu** [1,*] , **Changlong Wang** [2,*] **and Lu Wang** [2]

1   Science and Technology on Communication Information Security Control Laboratory, Jiaxing 314033, China
2   The Key Laboratory of Electronic Information Countermeasure and Simulation Technology Ministry of Education, Xidian University, Xi'an 710071, China
*   Correspondence: zhuhuijie@zju.edu.cn (H.Z.); clw_xjtu@163.com (C.W.)

**Abstract:** At present, there are many technical methods in the field of target detection, but the detection methods are greatly affected by direct-path signals and need technical support such as extracting pure direct-path signals, so they cannot be used under the condition of without-direct-path signals. In this paper, a distributed target detection method is studied under a without-direct-path signals system. In the case of without-direct-path signals, target detection is achieved by the generalized likelihood ratio test (GLRT). At the same time, the input echo signals in target detection need time synchronization, so the impact of time delay should be eliminated. However, the traditional time delay estimation method is realized through coherent processing between echoes, which requires a high signal-to-noise ratio (SNR). Therefore, a time delay estimation method is proposed in this paper. Finally, the experimental results show that the accuracy of target detection by GLRT is improved, and the signal-to-noise ratio is also improved under the condition of without-direct-path signals. Moreover, the accuracy of delay detection is improved.

**Keywords:** passive radar; target detection; without-direct-path signals; GLRT





## 1. Introduction

Passive radar refers to a radar detection system that does not emit electromagnetic wave signals but relies on existing electromagnetic waves in space to achieve detection, positioning, and tracking [1–5]. Passive radar can detect targets by receiving electromagnetic wave signals from third-party non-cooperative illuminators reflected by targets. Such third-party non-cooperative illuminators include commercial radio stations, television stations, communication base stations, and satellites. There are many types of signals emitted by external illuminators available. The passive radar system has the following advantages: high concealment, with strong anti-interference and anti-radiation missile capabilities; it has the potential of anti-stealth; the available external illuminators have abundant signals and strong networking capability; the system has low cost and no pollution to the environment; it can detect low altitude and ultra-low altitude targets. Because the passive radar system has many advantages, scholars at home and abroad have performed much research on target detection algorithms for passive radar.

At present, the cross-correlation method is mainly used for target detection. The cross-correlation detection method is based on the matched filtering theory, which uses the ambiguity function of the reference signal and the target echo signal to accumulate the target energy, obtaining the target delay and Doppler information so as to achieve target detection. The passive radar detection system obtains the reference signal by purifying the direct-path signals. The purity of the reference signal has an important impact on the system detection performance. The reference signal can eliminate the clutter in the echo signal and improve the signal-to-noise ratio of the target echo through matched filtering. Therefore, it is crucial for the passive radar to obtain pure direct-path signals for target

detection [6–10]. At present, scholars have performed much research on direct-path signal purification. T. Ying proposed a new direct-path signals purification method based on sparse characteristics for the problem of direct wave purification in non-cooperative passive detection [11] J. Wang proposed a direct wave purification method for passive radar in the scene with strong multipath signals reflected [12].

Although the target detection algorithm based on passive radar has gradually matured, there are still some limitations that limit the applicable environment of passive radar. The significance of direct-path signals target detection for passive radar is very significant. Therefore, if there is a scene where direct-path signals cannot be received, as shown in Figure 1. When each receiver cannot receive direct-path signals and can only receive the echo signals reflected from the target separately, the traditional cross-correlation method is no longer applicable, and the target detection performance will decline sharply.

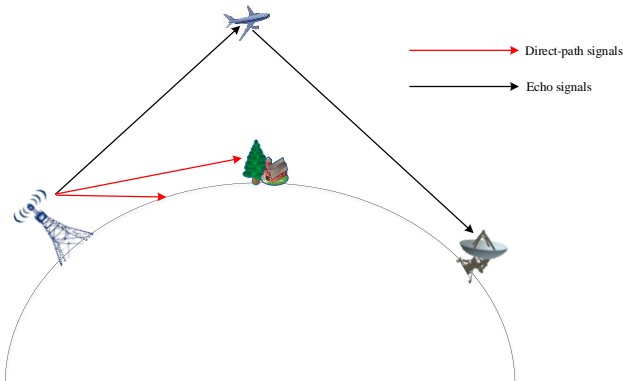

**Figure 1.** The condition of without-direct-path signals.

At present, experts have conducted some research on target detection algorithms based on the condition of without-direct-path signals and generally use the generalized likelihood ratio test (GLRT) detection method. Its core idea is to substitute the maximum likelihood estimates of all unknown parameters into the likelihood ratio to form a detector and compare them with the threshold to achieve target detection [13–23]. For a passive radar system with one illuminator and multiple receivers, Bialkowski, K.S. proposed a generalized canonical correlation (GCC) detector for passive multistate radar detection under the condition of known noise power [24] J. Liu proposed a generalized likelihood ratio test (GLRT) detector to deal with the target detection problem of passive radar under the condition of without-direct-path signals. Although it solves the detection problem of without-direct-path signal scenes to a certain extent, the improvement of SNR by these two detection algorithms is not very obvious [13].

In order to solve the problem of target detection when the direct-path signals cannot be received in practice, this paper proposes a distributed target detection method under the without-direct-path signals system based on the principle of GLRT. This paper first introduces the research status of passive radar and target detection of passive radar and then proposes the corresponding improvement methods. Then, the specific scene of without-direct-path signals is modeled, and the target detection algorithm based on the specific scene is described in detail. Finally, the above algorithms are simulated and processed with the measured data. The effectiveness of the algorithms is verified through simulation and further applied to the measured echo signals.

## 2. The Scene of Without-Direct-Path Signals

Based on modern detection theory, this paper studies the distributed target detection problem of passive radar based on without-direct-path signals, derives the generalized likelihood ratio test for this problem, determines the distribution of test statistics under two assumptions, and uses its distribution characteristics to achieve target detection. Its structure diagram is shown in Figure 2, including $N_r$ receivers and an illuminator. Each

receiver receives only the echo signals of the target. The location of the illumination source is indicated by **d**, the location of the $j$th receiver is indicated by $\mathbf{r}_j$, and the location of the target is indicated by **t**. The number of arrays of each receiver is $M$, the distance between the array is d, $\theta_{jk}$ is the direction of arrival relative to the $j$th receiver, then the difference of phase $\varphi_{jk}^{(i)}$ is given by

$$\varphi_{jk}^{(i)} = \frac{2\pi d \sin \theta_{jk}}{\lambda,}, i = 1, 2 \cdots, M. \tag{1}$$

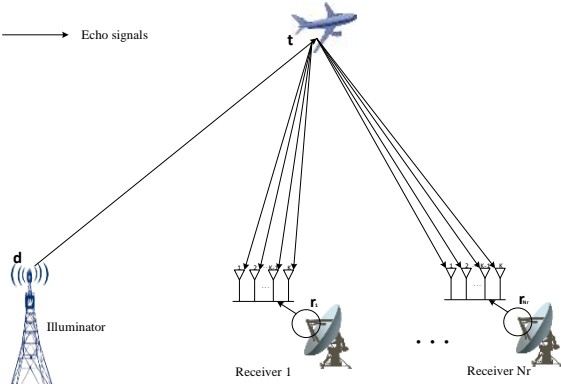

**Figure 2.** Passive radar geometric model.

Assume that the signal is independent of the noise. Then, the guide vector $\mathbf{a}_j(\theta_k)$ is given by

$$\mathbf{a}_j(\theta_k) = \left[ e^{j\varphi_{jk}^{(1)}}, e^{j\varphi_{jk}^{(2)}}, \cdots e^{j\varphi_{jk}^{(M-1)}}, e^{j\varphi_{jk}^{(M)}} \right]. \tag{2}$$

Suppose the target position is **t**, and let $R_1 = \|\mathbf{t} - \mathbf{d}\|$ and $R_{2j} = \|\mathbf{r}_j - \mathbf{t}\|$. Then the target-path propagation delay is $\tau_j = \left( R_1 + R_{2j} \right)/c$, $c$ is the speed of light. $v_j$ is the speed of the target. Similarly, $f_{dj} = 2v_j/\lambda$ is the target-path Doppler shift, where $\lambda = c/f_c$ is the wavelength of the transmitter, and $f_c$ is the carrier frequency of the transmitter.

And assuming that the signal accumulation time is $T$, the sampling rate is $f_s$, and the signal length is $L = Tf_s$, the signal received by the $k$th array in the $j$th receiver is $s_j^k$, given by

$$s_j^k = \gamma_j e^{i\varphi_{jk}^{(k)}} D_j \mathbf{u} + \mathbf{n}_j^k, s_j^k \in \mathbb{C}^{L*1}, \tag{3}$$

Where $\mathbf{u} \in \mathbb{C}^{L*1}$ is the complex baseband signal transmitted by the illuminator; $i$ is the imaginary unit; $D_j = D\left( \tau_j, f_{dj} \right) \in \mathbb{C}^{L*L}$ is a Doppler operator considering delay and Doppler shift. $\gamma_j$ is the channel coefficient, $\mathbf{n}_j^k$ is a complex Gaussian noise distribution of $CN(0, \sigma^2)$, where $\sigma^2$ unknown.

Then, concatenates the time series vector of all $M$ elements from the $j$th receiver

$$s_j = \gamma_j \left( D_j \otimes \mathbf{a}_j \right) \mathbf{u} + \mathbf{n}_j, s_j \in \mathbb{C}^{ML*1}, \tag{4}$$

where $\otimes$ is the Kronecker product, $\mathbf{n}_j \in \mathbb{C}^{ML*1}$, $\mathbf{a}_j \in \mathbb{C}^{M*1}$, $\mathbf{u} \in \mathbb{C}^{L*1}$, $s_j \in \mathbb{C}^{ML*1}$.

## 3. GLRT Detection Algorithm

### 3.1. Detection Algorithm

For the receivers of the passive radar, the waveform emitted by the non-cooperative source is generally unknown. Therefore, traditional matched filtering cannot be used to achieve object detection, and multiple receivers can be used to collect target echo signals from different directions to achieve object detection.

In passive radar systems, many radar systems require an additional independent channel as a reference channel, which is often used to collect direct-path signals from the irradiation source as a reference for target detection. Direct-path signals can be used to cross-correlate with echoes to achieve target detection, so obtaining a relatively pure direct-path signal is the key to achieving target detection. However, since the passive radar may not receive direct-path signals, the above method does not apply, so this paper adopts a target detection method based on without-direct-path signals.

In the radar system based on the passive radiation source, the passive radiation source will send an electromagnetic pulse if there is a target, the electromagnetic wave transmitted to the target will be transmitted to the receiver, and the target echo signal received by the radar system will contain various parameter information of the target. If the target is not present, the received signal contains only noise. Obviously, the presence or absence of the target echo waves constitutes two assumptions. According to two assumptions, a decision is made that the target signal is present or that only noise is present. The assumption $H_1$ indicates that the signal received by the receiving base station contains the target echo signals, and the assumption $H_0$ indicates that the signal received by the receiving base station does not contain the target echo, and the target detection problem becomes a binary hypothesis problem, as shown in Equation (5).

$$\begin{aligned} H_1 &: s_j = \gamma_j \left( D_j \otimes \mathbf{a}_j \right) \mathbf{u} + \mathbf{n}_j \\ H_0 &: s_j = \mathbf{n}_j \end{aligned} \tag{5}$$

Let $s = [s_1^T, \cdots, s_{N_r}^T] \in \mathbb{C}^{MN_rL*1}, \gamma = [\gamma_1 \cdots \gamma_{N_r}]^T \in \mathbb{C}^{N_r*1}$.

Due to the independence of receiver noise across the transmitter channels, for different receivers, it can be assumed that the received echo signals are independent of each other and that the echo signal is independent of the noise. The probability density function (PDF) under $H_1$ conditions can be written as $p_1(s|\gamma, \mathbf{u}), p_1(s|\gamma, \mathbf{u})$ as shown in Equation (6).

$$p_1(s|\gamma, \mathbf{u}) = \exp \left\{ -\frac{1}{\sigma^2} \sum_{j=1}^{N_r} \left\| s_j - \gamma_j \left( D_j \otimes \mathbf{a}_j \right) \mathbf{u} \right\|^2 \right\} \tag{6}$$

$\|\cdot\|$ represents the Euclidean norm. The probability density function (PDF) under $H_0$ condition can be written as $p_0(s), p_0(s)$ as shown in Equation (7).

$$p_0(s) = c_n \exp \left\{ -\frac{1}{\sigma^2} \left\| s \right\|^2 \right\} \tag{7}$$

where $c_n = \left( \pi \sigma^2 \right)^{-N_r L}$ is a normalization constant.

Because the transmitted signal and channel coefficients are non-deterministic and unknown, this optimal detector cannot be obtained directly. However, the actual detection algorithm can be designed according to the GLRT criterion, that is, to replace the unknown parameter with the maximum likelihood estimation (MLE).

Let $l_1(\gamma, \mathbf{u}|s) = \log p_1(s|\gamma, \mathbf{u})$ represent the log-likelihood function under $H_1$. Similarly, let $l_0(s) = \log p_0(s)$ represent the log-likelihood function under $H_0$, GLRT can be written as

$$\max_{\{\gamma, \mathbf{u}\}} l_1(\gamma, \mathbf{u}|s) - l_0(s) \underset{H_0}{\overset{H_1}{\gtrless}} \varsigma, \tag{8}$$

where $\varsigma$ is the threshold, ignoring the additive constant, which $l_1(\gamma, \mathbf{u}|s)$ can be written as

$$l_1(\gamma, \mathbf{u}|s) = -\frac{1}{\sigma^2} \sum_{j=1}^{N_r} \left\| s_j - \gamma_j \left( D_j \otimes \mathbf{a}_j \right) \mathbf{u} \right\|^2. \tag{9}$$

According to Equation (7), the maximum likelihood estimate (MLE) of $\gamma$ is

$$\hat{\gamma} = \frac{\left(\left(D_j \otimes \mathbf{a}_j\right)\mathbf{u}\right)^H s_j}{\left\|\left(D_j \otimes \mathbf{a}_j\right)\mathbf{u}\right\|^2}, \tag{10}$$

where $[\cdot]^H$ is the Hermitian transpose of $[\cdot]$, Equation (10) can be simplified to

$$\hat{\gamma} = \frac{\left(\left(D_j \otimes \mathbf{a}_j\right)\mathbf{u}\right)^H s_j}{\left\|\left(D_j \otimes \mathbf{a}_j\right)\mathbf{u}\right\|^2}, \tag{11}$$

where $\|\mathbf{u}\|^2 = L$, $\widetilde{s}_{sj} = \left(D_j\right)^H s_j$, $s_j$ is the echo signal received by the $j$th receiver, $\widetilde{s}_{sj}$ is the signal after removing the delay and Doppler shift, substitute $\gamma$ into Equation (6) can be obtained

$$l_1(\hat{\gamma}, \mathbf{u}|s) = -\frac{1}{\sigma^2}\left(\|s\|^2 - \frac{(\mathbf{u})^H \Phi(\Phi)^H \mathbf{u}}{\|\mathbf{u}\|^2}\right), \tag{12}$$

where $\Phi = [\widetilde{s}_{s1}, \cdots, \widetilde{s}_{sNr}] \in \mathbb{C}^{ML*N_r}$, let $\lambda_1(\cdot)$ denote the maximum eigenvalue of the matrix and let $\mathbf{v}_1(\cdot)$ denotes its associated eigenvector, then the maximization of Equation (10) is equivalent to maximizing the Rayleigh quotient, and when $\lambda_1(\Phi\Phi^H)$ taking and its associated eigenvector $\mathbf{u} = \mathbf{v}_1\left(\Phi\Phi^H\right)$, the Rayleigh quotient reaches its maximum value so $\hat{\mathbf{u}} = \mathbf{v}_1(\Phi\Phi^H)$ [25], so $l_1\left(\hat{\gamma}, \hat{\mathbf{u}}|s\right)$ can be written as

$$l_1\left(\hat{\gamma}, \hat{\mathbf{u}}|s\right) = -\frac{1}{\sigma^2}\|s\|^2 + \frac{1}{\sigma^2}\lambda_1\left(\Phi\Phi^H\right), \tag{13}$$

where $P = \Phi\Phi^H$, $\lambda_1(\Phi\Phi^H) = \lambda_1(\Phi^H\Phi)$, $N_r \ll L$, and for more efficient calculations, Equation (13) can be simplified to

$$l_1\left(\hat{\gamma}, \hat{\mathbf{u}}|s\right) = -\frac{1}{\sigma^2}\|s\|^2 + \frac{1}{\sigma^2}\lambda_1(P). \tag{14}$$

Similarly, under $H_0$ conditions

$$l_0(s) = -\frac{1}{\sigma^2}\|s\|^2. \tag{15}$$

Substituting Equations (14) and (15) into Equation (8) yields:

$$\xi = -\frac{1}{\sigma^2}\lambda_1(P) \underset{H_0}{\overset{H_1}{\gtrless}} \varsigma, \tag{16}$$

where $\varsigma$ is the detection threshold, which is derived from statistical averages.

The overall flow of the algorithm is shown in Figure 3.

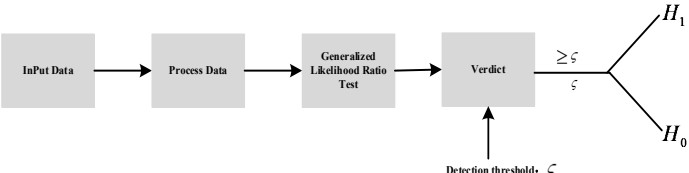

**Figure 3.** Flow chart of the target detection algorithm.

### 3.2. The Estimation of Time Delay of Arrival (TDOA)

Because $\widetilde{s}_{sj}$ is the signal after removing the delay and Doppler shift and the estimation of TDOA can serve as the basis for positioning So, a new method of TDOA estimation is proposed in this paper. The steps are as follows.

1. Generate the scheme of TDOA estimation;
2. Set the time delay interval $b$, according to the distance between stations, set the search range as $[\tau_{\min}, \tau_{\max}]$, and set the scheme $S$. There are $N_r - 1$ auxiliary receivers, easy to know $S$ is a finite collection. The number of elements is $(\tau_{\max} - \tau_{\min} + 1)^{N_r - 1}$, and $S$ can be expressed as

$$S = \{x \in R^{N_r - 1} | x_i \in N, \tau_{\min} \leq x_i \leq \tau_{\max}, i = 1, 2, \cdots, N_r - 1\};$$
$$= \left\{S_1, S_2, S_3, \cdots, S_{(\tau_{\max} - \tau_{\min} + 1)^{N_r - 1}}\right\}$$

3. For any $x^* = (x_1, x_2, \cdots, x_{N_r - 1}) \in S$, represents the $j$th echo data from the receivers moving to the right $x_{j-1} \cdot b$ time units;
4. Initialize the $1 \times (\tau_{\max} - \tau_{\min} + 1)^{N_r - 1}$ dimensional vector $G = [g_1, g_2, g_3, \cdots, g_{(\tau_{\max} - \tau_{\min} + 1)^{N_r - 1}}]$ and set all the entries in $G$ to 0;
5. Estimate the TDOA according to the scheme;
6. Go through all the schemes in the set $S$, and repeat the following (2) and (3) for each scheme until the collection $S$ is traversed and step 3 is entered;
7. The echo data of each channel was arranged as $1 \times L$ dimensional vector in chronological order; that is $\left(s_j^k\right)^T \in \mathbb{C}^{1*L}$, the echo data of $N_r$ channel was merged into a $N_r * L$ dimensional matrix $\mathbf{X}$;
8. Make a calculation of

$$\xi = -\frac{1}{\sigma^2}\lambda_1(\mathbf{X}\mathbf{X}^H)$$

9. Put $\xi$ in the corresponding position of vector $G$, let us say $g_i = a, 1 \leq i \leq (\tau_{\max} - \tau_{\min} + 1)^{N_r - 1}$;
10. Traverse the vector $G$ to find the maximum value, as shown in (17);

$$z = \max_{i \in \{1, 2, \cdots, (\tau_{\max} - \tau_{\min} + 1)^{N_r - 1}\}} G_i \tag{17}$$

11. Record the position of the maximum value in $G$, and find out the corresponding scheme of the same position as in $G$ and $S$, so as to determine the delay time $\tau_0$, as shown in Equation (18).

$$\tau_0 = \underset{i \in \{1, 2, \cdots, (2M+1)^{K-1}\}}{\arg\max} G_i \tag{18}$$

## 4. Experiments

In this part, the signal of DVB-T is used to verify the performance of target detection and the accuracy of TDOA estimation.

### 4.1. Detection Performance

The type of signal is DVB-T. The mode of DVB-T is 2 k, the mode of symbol mapping of DVB-T is QPSK, and the bandwidth of DVB-T is 8 MHz. Then, the echo signals are generated based on the target speed, position, and distance between the receivers and the SNR.

In order to verify the detection performance designed in this paper, this part performs simulation experiments on the previously designed detection algorithm and gives simulation results. Table 1 shows the parameter settings used.

**Table 1.** Simulation parameter settings of target detection.

| Parameters | Value |
| --- | --- |
| Number of receivers | 4 |
| Sampling frequency | 100 MHz |
| Destination location | (0, 50, 0) km |
| Receivers position | (10, 0, 0) km, (−10, 0, 0) km<br>(0, 10, 0) km, (0, −10, 0) km |
| Illuminator location | (0, 0, 0) km |
| Target speed | 0 |
| Signal length | 0.01 s |
| SNR | −40 dB |
| Type of signal | DVB-T |

According to the parameters set in Table 1. The original signal was normalized, and then the experiment was conducted. A total of 1000 Monte Carlo experiments were conducted. The Monte Carlo experiment results are shown in Figure 4. The original signal "1" indicates echo signal, and "0" indicates no echo signal.

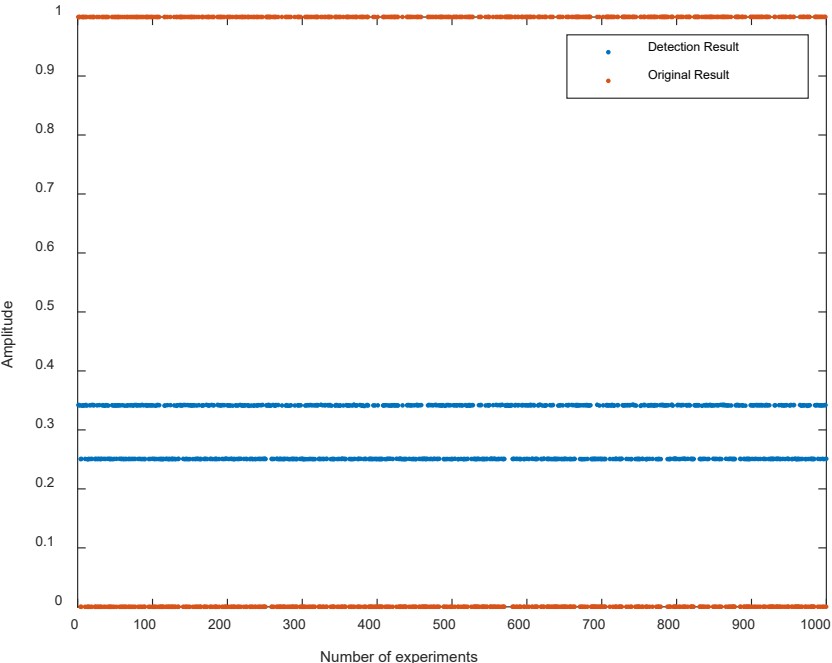

**Figure 4.** Detection result.

The results of 1000 Monte Carlo experiments are shown in Figure 4. The experimental results show that when there are echo signals, the value of $\xi$ is about 0.34. When there is no echo signal, the value of $\xi$ is about 0.25. At this time, the detection threshold is 0.3 on average. When it is greater than the threshold, an echo signal is present. When it is less than the threshold, only a noise signal is present. The original results are consistent with the detection results, and the detection accuracy is 100%.

Next, the relationship between target detection performance and SNR and signal length is verified. Each table lists the relationship between SNR and detection probability (DP). The SNR is set as −39 dB~−10 dB, the interval of each SNR change is 1, and the signal length is set as 1 ms, 10 ms, 0.1 s, and 1 s, respectively. The results are shown in Tables 2–5 and Figure 5.

**Table 2.** Signal length: 1 ms.

| SNR/dB | −10 | −11 | −12 | −13 | −14 | −15 | −16 | −17 | −18 | −19 |
|---|---|---|---|---|---|---|---|---|---|---|
| DP | 1 | 1 | 1 | 1 | 1 | 0.999 | 0.98 | 0.929 | 0.842 | 0.722 |
| SNR/dB | −20 | −21 | −22 | −23 | −24 | −25 | −26 | −27 | −28 | −29 |
| DP | 0.639 | 0.549 | 0.479 | 0.456 | 0.429 | 0.426 | 0.407 | 0.427 | 0.407 | 0.39 |
| SNR/dB | −30 | −31 | −32 | −33 | −34 | −35 | −36 | −37 | −38 | −39 |
| DP | 0.423 | 0.408 | 0.382 | 0.398 | 0.392 | 0.409 | 0.433 | 0.396 | 0.399 | 0.403 |

**Table 3.** Signal length: 10 ms.

| SNR/dB | −10 | −11 | −12 | −13 | −14 | −15 | −16 | −17 | −18 | −19 |
|---|---|---|---|---|---|---|---|---|---|---|
| DP | 1 | 1 | 1 | 1 | 1 | 1 | 1 | 1 | 1 | 0.999 |
| SNR/dB | −20 | −21 | −22 | −23 | −24 | −25 | −26 | −27 | −28 | −29 |
| DP | 0.995 | 0.987 | 0.94 | 0.851 | 0.724 | 0.649 | 0.553 | 0.5 | 0.439 | 0.444 |
| SNR/dB | −30 | −31 | −32 | −33 | −34 | −35 | −36 | −37 | −38 | −39 |
| DP | 0.414 | 0.464 | 0.427 | 0.412 | 0.401 | 0.405 | 0.402 | 0.407 | 0.378 | 0.418 |

**Table 4.** Signal length: 100 ms.

| SNR/dB | −10 | −11 | −12 | −13 | −14 | −15 | −16 | −17 | −18 | −19 |
|---|---|---|---|---|---|---|---|---|---|---|
| DP | 1 | 1 | 1 | 1 | 1 | 1 | 1 | 1 | 1 | 1 |
| SNR/dB | −20 | −21 | −22 | −23 | −24 | −25 | −26 | −27 | −28 | −29 |
| DP | 1 | 1 | 1 | 1 | 1 | 0.997 | 0.984 | 0.956 | 0.845 | 0.762 |
| SNR/dB | −30 | −31 | −32 | −33 | −34 | −35 | −36 | −37 | −38 | −39 |
| DP | 0.661 | 0.535 | 0.497 | 0.476 | 0.435 | 0.422 | 0.419 | 0.455 | 0.411 | 0.414 |

**Table 5.** Signal length: 1 s.

| SNR/dB | −10 | −11 | −12 | −13 | −14 | −15 | −16 | −17 | −18 | −19 |
|---|---|---|---|---|---|---|---|---|---|---|
| DP | 1 | 1 | 1 | 1 | 1 | 1 | 1 | 1 | 1 | 1 |
| SNR/dB | −20 | −21 | −22 | −23 | −24 | −25 | −26 | −27 | −28 | −29 |
| DP | 1 | 1 | 1 | 1 | 1 | 1 | 1 | 1 | 1 | 1 |
| SNR/dB | −30 | −31 | −32 | −33 | −34 | −35 | −36 | −37 | −38 | −39 |
| DP | 1 | 0.96 | 0.91 | 0.83 | 0.81 | 0.7 | 0.419 | 0.63 | 0.57 | 0.57 |

Monte Carlo experiments were conducted 1000 times, and the results of Monte Carlo experiments are shown in Figure 5. With the increase in SNR, the detection probability increased continuously. Under the condition of high SNR, the detection probability could reach 1. When the accumulation time is 1 ms, the SNR above −20 dB can achieve better detection results. When the accumulation time is 10 ms, the SNR above −25 dB can achieve better detection results. When the accumulation time is 100 ms, the SNR above −30 dB can achieve better detection results. When the accumulation time is 1 s, the SNR above −35 dB can achieve better detection results. The results show that with the increase in signal length, the detection effect is significantly improved.

### 4.2. Verify the Accuracy of TDOA Estimation

In order to verify the estimation algorithm designed in this paper, this part carries out simulation experiments on the previous estimation algorithm and gives the simulation results. The following table (Table 6) lists the parameters to be set.

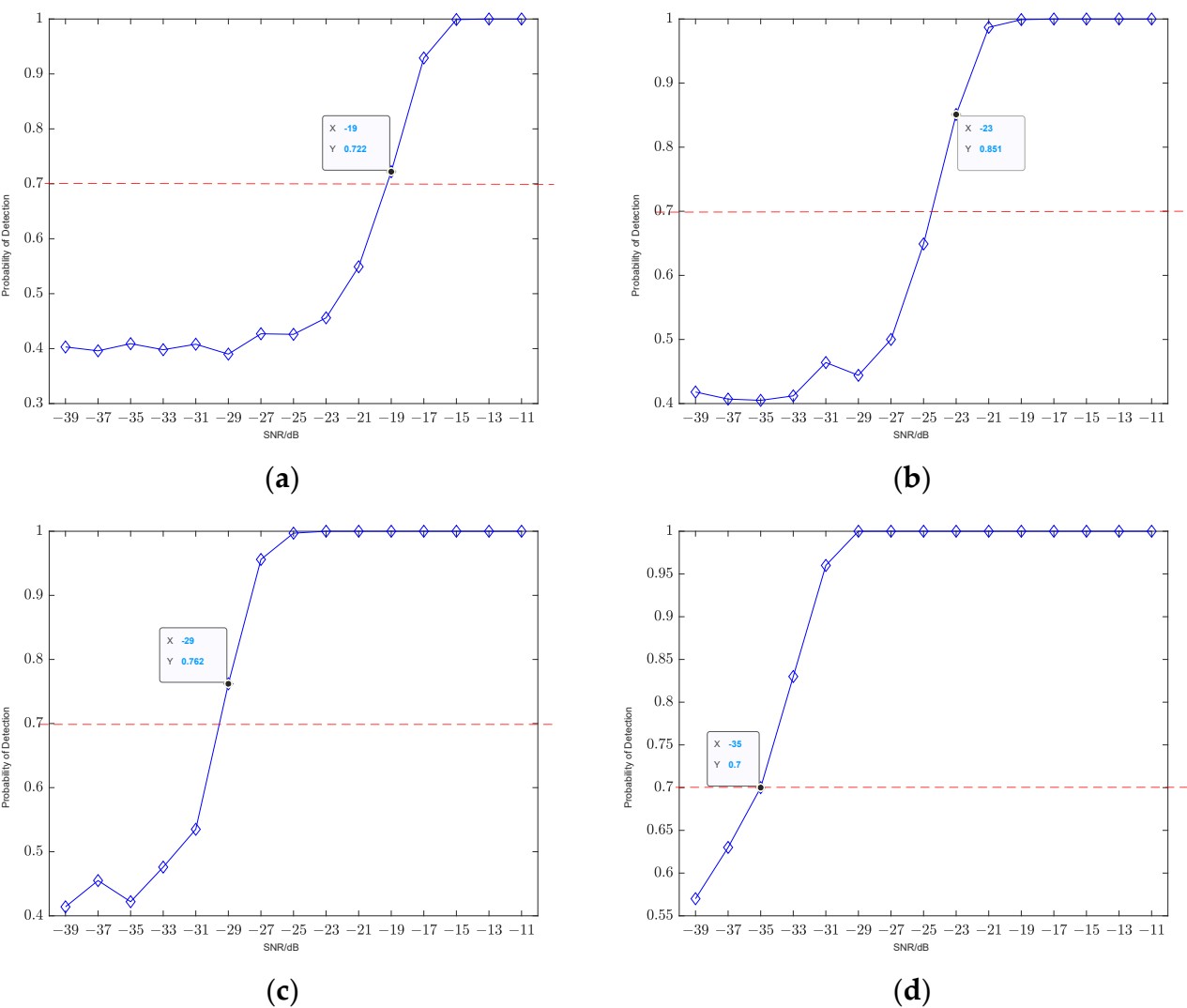

**Figure 5.** The relationship between detection probability and SNR and signal length. (**a**) The signal length is 1 ms; (**b**) the signal length is 10 ms; (**c**) the signal length is 100 ms; (**d**) the signal length is 1 s.

**Table 6.** The parameter of TDOA estimation.

| Parameters | Value |
|---|---|
| Number of receivers | 2 |
| SNR | $-40$ dB |
| Signal length | 0.1 s |
| Sampling frequency | 100 MHz |
| $[\tau_{\min}, \tau_{\max}]$ | $[-0.1 \text{ ms}, 0.1 \text{ ms}]$ |
| Time moving interval $b$ | 0.001 ms |
| Target speed | 0 |
| Receivers position | $(10, 0, 0)$ km, $(-10, 0, 0)$ km |
| Illuminator location | $(0, 0, 0)$ km |

Different time delays were set, including $-0.08$ ms, $-0.03$ ms, 0 ms, 0.03 ms, 0.04 ms, and 0.05 ms. Monte Carlo experiments were conducted 1000 times at each time delay to verify the correctness, and the results are shown in Figure 6.

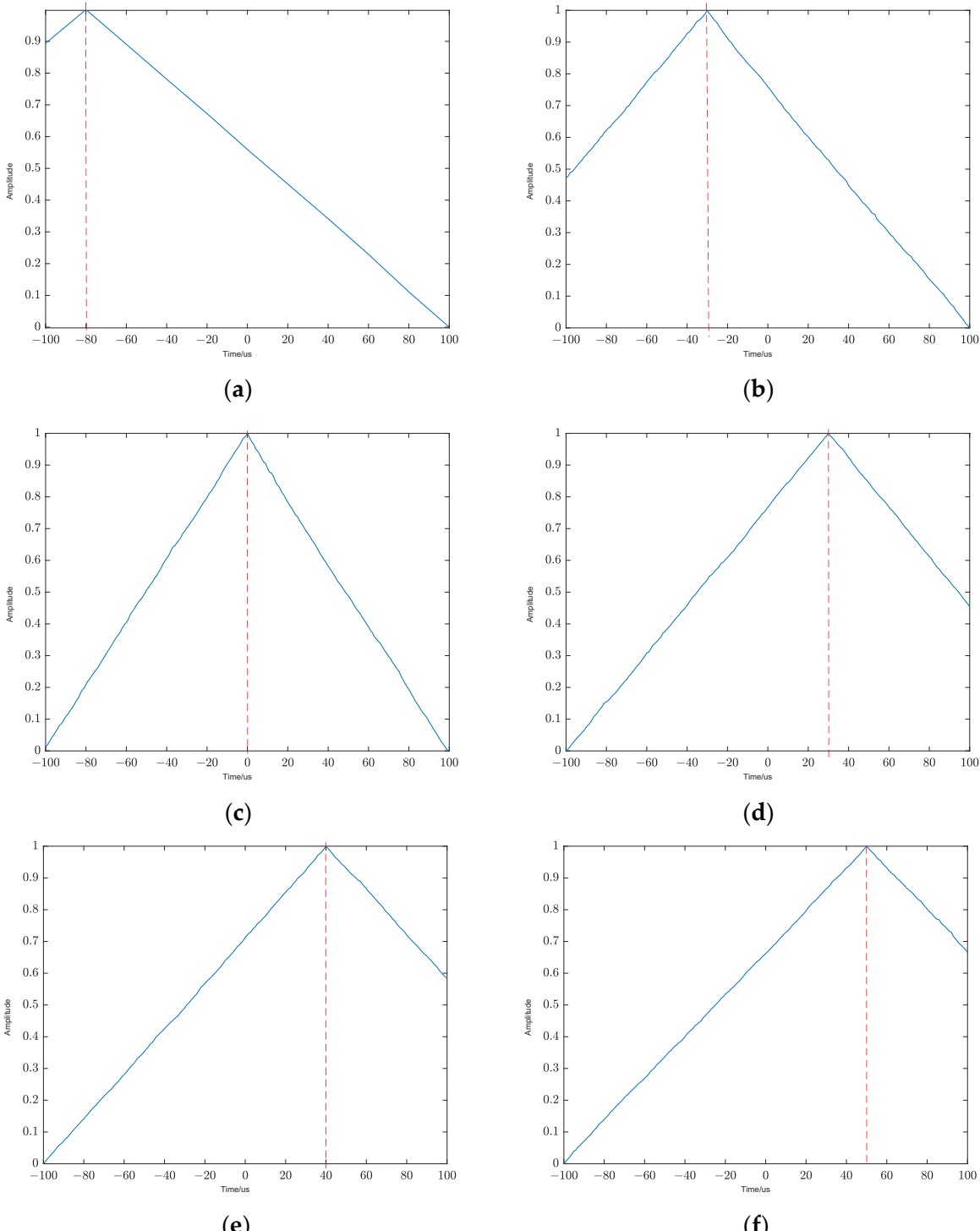

**Figure 6.** The result of different TDOA estimation results. (**a**) The TDOA is −0.08 ms; (**b**) the TDOA is −0.03 ms; (**c**) the TDOA is 0 ms; (**d**) the TDOA is 0.03 ms; (**e**) the TDOA is 0.04 ms; (**f**) the TDOA is 0.05 ms.

The experimental results are shown in Figure 6. It can be seen that under −40 dB, the delay at the detection peak is consistent with the set delay, and the delay detection accuracy reaches 100%.

Next, the relationship between delay estimation and SNR is verified. The SNR is −50~−31 dB, and the signal accumulation time is 0.1 s. The experimental results are shown in Figure 7.

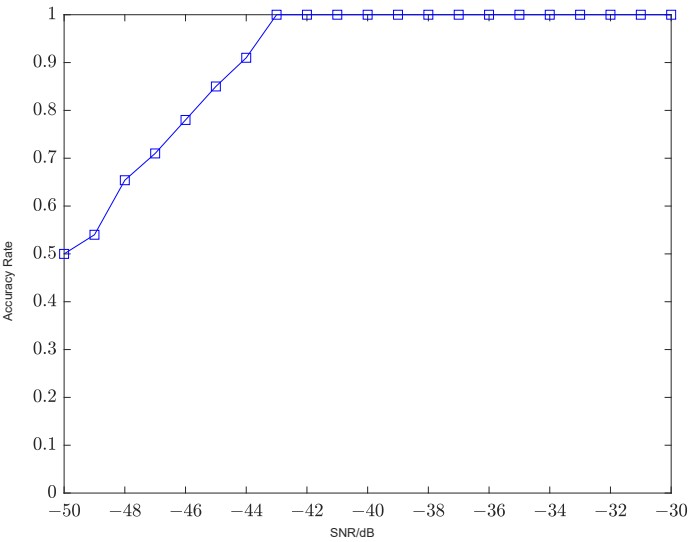

**Figure 7.** Delay detection accuracy.

As can be seen in Figure 7, when −50~−31 dB, the accuracy of TDOA estimation is 1 when the SNR is above −35 dB, indicating that the method of TDOA estimation in this paper is insensitive to SNR and can adapt to more situations and lower SNR.

### 4.3. The Influence of Different Algorithms on Detection Performance

This part mainly verifies the detection algorithms of [26,27] and compares the performance of the algorithms proposed in this paper. The following table (Table 7) lists the parameters to be set.

**Table 7.** The parameter of detection scene.

| Parameters | Value |
| --- | --- |
| The location of illuminator | (0, 0, 0) km |
| The location of receiver | (0, 10, 0) km, (0, −10, 0) km |
| The location of target | (0, 50, 0.1) km |
| Temperature | 290 K |
| Bandwidth | 10 MHz |
| The gain of transmitter | 10 dB |
| The gain of receiver | 10 dB |
| Type of signal | DVB-T |

According to the scene set in the table above, then set parameters: radar cross section (RCS) and the power of the transmitter. Change the RCS from 10 m$^2$ to 500 m$^2$, and change the power of the transmitter from 1000 W to 30,000 W, so that the SNR varies in the range of −40 dB to −10 dB so as to verify the detection probabilities of different detection methods under different SNR. The authors of [26] use the GLRT method to detect targets in the presence of direct waves; [27] use the blind channel estimation method for target detection. The experimental results are shown in Figure 8.

It can be seen from Figure 8 that the detection performance of the method proposed in this paper is the best. The performance of the proposed method and the method of [26] is better when SNR is above −30 dB, but the detection performance decreases rapidly when SNR is below −30 db. However, the blind channel estimation method in [27] has poor performance at low SNR.

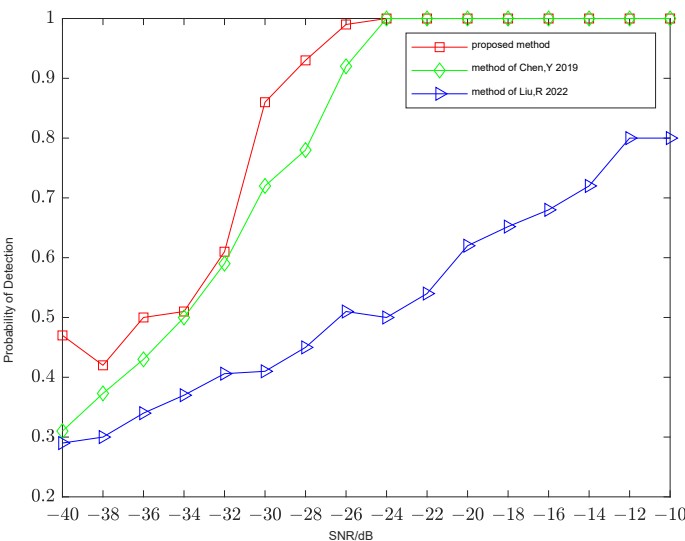

**Figure 8.** Performance comparison of different algorithms.

According to the calculation, the TDOA between two receivers is −67 us. The algorithm proposed in this paper can be used to correctly estimate the TDOA, and the results are shown in Figure 9.

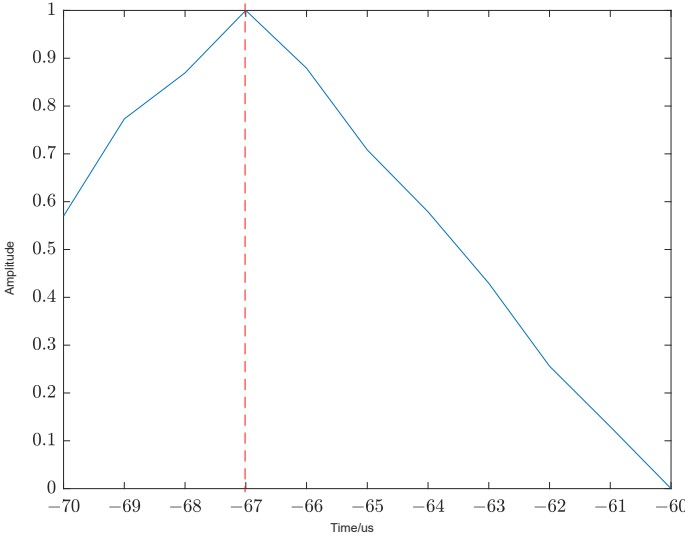

**Figure 9.** The result of TDOA estimation.

As can be seen from Figure 9, after normalization, the peak value is at −67 us, which is consistent with the calculation result, indicating that the TDOA estimation result is correct. The algorithm proposed in this paper can not only achieve target detection but also extract the delay parameters.

## 5. Conclusions

This paper focuses on a distributed target detection method in the without-direct-path signals system. In the case of without-direct-path signals, the target detection is realized by the GLRT. The relationship between the target detection performance, SNR, and signal length is verified. The higher the SNR, the longer the signal length and the better the target detection performance. The experimental results show that in the case of without-direct-path signals, the accuracy of target detection by GLRT is improved, and the delay detection algorithm in this paper is insensitive to the change of SNR, so the accuracy can reach 100% at −40 dB.

**Author Contributions:** Conceptualization, C.W. and H.Z.; methodology, C.W. and H.Z.; software, C.W. and L.W.; validation, H.Z., C.W. and L.W.; formal analysis, C.W.; investigation, L.W.; resources, C.W. and H.Z.; data curation, H.Z.; writing—original draft preparation, L.W.; writing—review and editing, C.W. and H.Z.; visualization, L.W.; supervision, C.W.; project administration, H.Z.; funding acquisition, H.Z. and C.W. All authors have read and agreed to the published version of the manuscript.

**Funding:** This research was funded in part by the China Postdoctoral Science Foundation, grant nos. 2020M673346, 2017M613076, and 2016M602775; in part by the National Natural Science Foundation of China, grant nos. 61801344, 61801347, 61631019, 61201418, and 62001350; in part by the Fundamental Research Funds for the Central Universities, grant nos. XJS200212, XJS200210, XJS200204, XJS210210, and JB210202; in part by the Aeronautical Science Foundation of China, grant no. 20181081003; and by the Postdoctoral Science Research Projects of Shaanxi Province, grant no. 2018 BSHEDZZ39.

**Data Availability Statement:** Not applicable.

**Conflicts of Interest:** The authors declare no conflict of interest. The funders had no role in the design of the study; in the collection, analyses, or interpretation of data; in the writing of the manuscript; or in the decision to publish the results.

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
