# Peer review of "A Target Detection Method of Distributed Passive Radar without Direct-Path Signal"

_electronics, doi:10.3390/electronics12020433_

Round 1

Reviewer 1 Report

1) GLRT algorithm is already proposed. What is the novel things you proposed?

2) References are very less. Use relevant references to make introduction part more technically strong.

3) What the variables and/or constants in the related equation represent should be stated after the equation.

4) How the detection probability can be justified based on the proposed approach? Also prove that the proposed algorithm works better compared to previously reported algorithm for same.

5) There are mainly three parameters are used to identify the novelty are; SNR, Signal length and relationship between the target detection performances. Kindly add another one or two parameters which can strengthen the result part.

6)In table 2 and 3 , what is PD?

7) What % of the echo signal is received without direct path method compared to indirect path method?

What the variables and/or constants in the related equation represent should be stated after the equation.

Author Response

A Target Detection Method of Distributed Passive Radar Without Direct-Path Signal

Huijie Zhu 1, Changlong Wang 2 and Lu Wang 2

Authors’ Reply to Reviewers’ Comments

Original Manuscript ID: electronics-2133643

Original Article Title: “A Target Detection Method of Distributed Passive Radar Based on Without Direct-Path Signal

General Comments: We would like to thank the editors and reviewers for their excellent comments and for taking the time to consider our paper. We have tried our best to revise the paper in accordance with their comments.

-------------------------------------------------------------------------------------------------------

Reviewers' comments
Reviewer #1:

1) GLRT algorithm is already proposed. What is the novel things you proposed?

Answer:The traditional GLRT algorithm is mainly applied in the case of direct wave. This paper mainly aims at the scene without direct wave, and the proposed algorithm can not only judge whether there is a target, but also extract the time difference parameter, which lays a certain foundation for positioning.

2) References are very less. Use relevant references to make introduction part more technically strong.

Answer:  More references have been added to the paper

3) What the variables and/or constants in the related equation represent should be stated after the equation.

Answer: Related variables have been explained at the end of the equation in the paper.

4) How the detection probability can be justified based on the proposed approach? Also prove that the proposed algorithm works better compared to previously reported algorithm for same.

Answer: The detection probability is verified under different signal-to-noise ratio and different signal length by simulation.

To add the comparative experiments, compared with the methods proposed in the two papers, "New Approximate Distributions for the Generalized Likelihood Ratio Test Detection in Passive Radar" and "Multi-target Detection by Distributed Passive Radar Systems without Reference Signals". Experimental results show that the proposed method is superior to the two methods.

5) There are mainly three parameters are used to identify the novelty are; SNR, Signal length and relationship between the target detection performances. Kindly add another one or two parameters which can strengthen the result part.

Answer: In the comparison experiment, variables RCS and transmitting power are added to verify their relationship with detection probability.

To add the comparative experiments, compared with the methods proposed in the two papers, "New Approximate Distributions for the Generalized Likelihood Ratio Test Detection in Passive Radar" and "Multi-target Detection by Distributed Passive Radar Systems without Reference Signals".

6)In table 2 and 3 , what is PD?

Answer: Has been changed to DP, where DP is the detection probability.

7) What % of the echo signal is received without direct path method compared to indirect path method?

Answer: Under certain circumstances, such as the curvature of the earth, the direct wave will not be received and so on.

Reviewer 2 Report

Summary:

There are various methods for target detection, but they rely on direct-path signals and require technical support to extract pure direct-path signals. Therefore, they cannot be used when direct-path signals are not available. This paper proposes a distributed target detection method that uses the Generalized Likelihood Ratio Test (GLRT) in the absence of direct-path signals. The input echo signals also need to be time synchronized, so the impact of time delay must be eliminated. The traditional method for estimating time delay involves coherent processing between echoes, which requires a high signal-to-noise ratio. The proposed method for time delay estimation does not have this requirement. The experimental results demonstrate that the accuracy of target detection using GLRT is improved and the signal-to-noise ratio is also improved in the absence of direct-path signals, and the accuracy of delay detection is improved.

Pros:

The GLRT method can be used for target detection in situations where direct-path signals are not available, which may be useful in a variety of settings.

The method does not require technical support to extract pure direct-path signals, which can make it simpler to use.

The proposed method for time delay estimation does not have the high signal-to-noise ratio requirement of traditional methods, which may make it more practical to use in some cases.

The experimental results demonstrate that the GLRT method can improve the accuracy of target detection, signal-to-noise ratio, and delay detection, which can be beneficial for various applications.

The method may be relatively simple to implement and use, depending on the specific application.

Cons:

One potential con of the method is that it has not been compared to previous methods that also use the GLRT or that can work without direct path signals, such as "New Approximate Distributions for the Generalized Likelihood Ratio Test Detection in Passive Radar" and "Multi-target Detection by Distributed Passive Radar Systems without Reference Signals". This may make it difficult to accurately evaluate the performance of the method and compare it to other approaches.

Author Response

A Target Detection Method of Distributed Passive Radar Without Direct-Path Signal

Huijie Zhu 1, Changlong Wang 2 and Lu Wang 2

Authors’ Reply to Reviewers’ Comments

Original Manuscript ID: electronics-2133643

Original Article Title: “A Target Detection Method of Distributed Passive Radar Based on Without Direct-Path Signal

General Comments: We would like to thank the editors and reviewers for their excellent comments and for taking the time to consider our paper. We have tried our best to revise the paper in accordance with their comments.

-------------------------------------------------------------------------------------------------------

Reviewers' comments
Reviewer #2:

One potential con of the method is that it has not been compared to previous methods that also use the GLRT or that can work without direct path signals, such as "New Approximate Distributions for the Generalized Likelihood Ratio Test Detection in Passive Radar" and "Multi-target Detection by Distributed Passive Radar Systems without Reference Signals". This may make it difficult to accurately evaluate the performance of the method and compare it to other approaches.

Answer: The comparative experiments have been supplemented in the paper. Experimental results show that the proposed method is superior to the two methods.
